# Age- and Lifespan-Dependent Differences in GO Caused DNA Damage in *Acheta domesticus*

**DOI:** 10.3390/ijms24010290

**Published:** 2022-12-24

**Authors:** Barbara Flasz, Marta Dziewięcka, Amrendra K. Ajay, Monika Tarnawska, Agnieszka Babczyńska, Andrzej Kędziorski, Łukasz Napora-Rutkowski, Patrycja Ziętara, Ewa Świerczek, Maria Augustyniak

**Affiliations:** 1Institute of Biology, Biotechnology and Environmental Protection, University of Silesia in Katowice, 40-007 Katowice, Poland; 2Department of Medicine, Division of Renal Medicine, Brigham and Women’s Hospital, Harvard Medical School, Boston, MA 02115, USA; 3Polish Academy of Sciences, Institute of Ichthyobiology and Aquaculture in Gołysz, 43-520 Chybie, Poland

**Keywords:** total DNA damage, DSB, 8-OHdG, AP sites, pATM, pH2A.X, Global DNA methylation, insects, carbon nanoparticles

## Abstract

The rising applicability of graphene oxide (GO) should be preceded by detailed tests confirming its safety and lack of toxicity. Sensitivity to GO of immature, or with different survival strategy, individuals has not been studied so far. Therefore, in the present research, we focused on the GO genotoxic effects, examining selected parameters of DNA damage (total DNA damage, double-strand breaks—DSB, 8-hydroxy-2′-deoxyguanosine-8-OHdG, abasic site—AP sites), DNA damage response parameters, and global methylation in the model organism *Acheta domesticus*. Special attention was paid to various life stages and lifespans, using wild (H), and selected for longevity (D) strains. DNA damage was significantly affected by stage and/or strain and GO exposure. Larvae and young imago were generally more sensitive than adults, revealing more severe DNA damage. Especially in the earlier life stages, the D strain reacted more intensely/inversely than the H strain. In contrast, DNA damage response parameters were not significantly related to stage and/or strain and GO exposure. Stage-dependent DNA damage, especially DSB and 8-OHdG, with the simultaneous lack or subtle activation of DNA damage response parameters, may result from the general life strategy of insects. Predominantly fast-living and fast-breeding organisms can minimize energy-demanding repair mechanisms.

## 1. Introduction

The last decades have brought about the dynamic development of production, characteristics, and description of the potential use of carbon nanoparticles and nanomaterials. Among them, graphene and graphene oxide (GO), probably the most often studied carbon nanostructures worldwide, seem very promising both in medical as well as numerous industrial applications [1,2,3,4]. GO, as well as its composites, can be used in energy storage, water and air purification, production of high-temperature materials, building materials, and electrodes [4,5]. In medicine, the use of GO relates to its use in drug delivery, biosensing and bioimaging, cancer and gene therapy, bone and teeth implants, scaffolds for cell culture, antimicrobial treatment, and many others [3,6,7].

In its 2D structure, GO, like graphene, contains carbon atoms linked in the shape of a honeycomb. In addition, oxygen functional groups (=O, -OH, -O-, -COOH) are attached to the GO surface and its edges. Their variable amount (from 3 to 40% oxygen in GO mass) of GO surface, resulting from the production method used, strongly determines the GO properties. An essential feature of GO is its hydrophilicity and the ability to form aqueous suspensions, which distinguishes this material from graphene [4,5,8,9]. These two features, the hydrophilicity and the high content of oxygen groups, allow GO to interact with biological/organic structures—penetrate organisms, organs, tissues, and cells and interact with molecules, including DNA (directly or indirectly). This phenomenon creates excellent opportunities for using GO in medicine and biology. However, it is also a cause for concern due to potentially adverse/uncontrolled interactions inside organisms/cells. Therefore, understanding the toxicity of GOs down to the last detail, especially in the in vivo model, and taking into account the condition and state of the organism, should not be underestimated. Each GO application, but especially the medical one, must guarantee the desired effect, but also a certainty that there will be minimal or no side effects from the treatment.

The toxicity of GO has been found in various cell lines and organisms, including bacteria, plants, and animals, with different life strategies [10,11,12,13,14,15,16,17,18]. In general, the adverse effects caused by GO included: increased mortality, developmental and/or reproductive disorders, weight loss, damage to the gut epithelium and other tissues, changes in the value of markers of inflammation and/or oxidative stress (increased generation of reactive oxygen species—ROS) [13,14,15,19,20,21,22]. Other concepts, postulated mainly to bacteria, are mechanical damage to membranes by sharp edges of GO flakes (nanoknives), wrapping or trapping of cells by GO film, and imbalance of lipid composition in the cell membrane [19,23,24]. Although many concepts and indirect evidence point to increased oxidative stress, the exact mechanism of GO toxicity still remains controversial. Scientists focus on the relationship between GO toxicity and the particles’ physical properties, dose, and exposure time [25]. Meanwhile, the sensitivity of a particular species to GO at various developmental stages (life stages) is practically not discussed. The specific burden of oxidative stress, which may be different for different stages of life, as well as improved/enhanced antioxidant mechanisms (resulting, for example, from selection), give rise to the hypothesis that GO (in given doses and time of exposure) may cause different effects at particular life stages of the same species. Special attention should be paid to a molecule of strategic importance to the individual and species—DNA.

As Wu et al. (2021) emphasized in the latest review on the effects of the graphene family nanomaterials (GFNs) on DNA, the genotoxicity mechanism of these materials, including GOs, is largely unknown [25]. The importance of monitoring substances with genotoxic properties has prompted the development and use of a few universal molecular biomarkers. These comprise the quantitative assessment of DNA damage (total DNA damage), including single (SSB) and double (DSB) DNA breaks, and evaluation of 8-hydroxy-2′-deoxyguanosine (8-OHdG) and apurinic/apyrimidinic sites (AP sites). Moreover, since DNA damage is always accompanied by repair, signaling pathway elements e.g., ATM kinase and histone H2A.X phosphorylation, are also frequently studied [26,27,28,29]. DNA damage and repair can vary at different ages. Therefore, data on global DNA methylation may provide additional information about the dynamics of the processes studied [25,30].

This study was aimed to evaluate DNA damage and elements of the repair pathway after using low doses of GO, with particular emphasis on different stages of life and potentially different coping capabilities with oxidative stress (resulting from multi-generational selection for longevity and different survival strategies). To achieve this goal, we used a unique strain of the selected model organism *Acheta domesticus*, which is characterized by several modifications indicating that the extension of life is accompanied by a shift in the redox balance [31,32,33,34].

Due to the potentially great application of GO in medicine, as well as the high probability of using it to produce antibacterial coatings and packaging (and thus the possibility of GO getting into the digestive tract), in this study, we were interested in relatively low (approaching real) doses of GO. Thus, compared to our previous studies [13,14,15,16], we lowered the GO doses by one hundred- to one thousand-fold. After one-generation exposure to GO at concentrations of 0.2 and 0.02 mg∙kg^−1^ of food, we measured the level of DNA damage (total DNA damage, DSB), the level of 8-OHdG and AP sites, and elements of the signaling pathway (ATM and H2A.X), as well as the global DNA methylation in larvae, young imago, and mature imago of *A. domesticus* from the wild and long-living strains. We found strain- and age-dependent DNA damage. DNA breaks and oxidative damage were more common in larvae and young imago, probably associated with intensive growth (larvae) or reproductive activity (young imago). The antioxidant profile developed over selection is also essential, and determines the manner and size of the response to GO. Therefore, in the future, attention should be paid to the problem of chronic, multigenerational exposure, and the level of methylation of critical genes involved in DNA repair and oxidative stress control.

## 2. Results

Firstly, ANOVA/MANOVA analysis for each main factor was performed separately, including the effects of strain (wild and long-living insects), treatment (control and receiving GO groups), and stage (developmental stages of *A. domesticus*—larvae, young imago, and mature imago). Then, statistical interactions between these factors were examined, i.e., their simultaneous impact on the dependent variables. The analysis showed that all the main effects and their interactions were significant. However, the strain × treatment and treatment × stage interactions were insignificant (Table 1).

Analysis of the main effects for the DNA repair path elements (pATM and pH2A.X parameters) showed no significant influence of strain, treatment, and stage factors. However, for the pH2A.X parameter, the treatment as well as stage effects were at the border of statistical significance. Additionally, for this parameter, significant interactions were revealed for the factors strain × treatment and strain × stage (Table 2).

### 2.1. DNA Damage: Total

The exposure of insects to GO, over one generation, caused DNA damage in the cells of the digestive tract. Total DNA damage (Figure 1) depended on stage and strain. A comparison of research groups showed that larva and young imago presented greater total DNA damage than mature imago. Although the statistical differences in larva groups from the H strain were visible only in the GO 0.2 group, when compared to the control, it was clear that GO-exposed larvae from this strain had a higher percentage of damaged DNA than the control group. Similar trends were observed for long-living strain (D) larvae. However, no significant differences in GO-treated larvae compared to control appeared. In young imago groups, wild-type insects presented total DNA damage on a comparable level. On the other hand, in the D strain, statistical differences were evident, showing the higher the GO concentration was, the higher the percentage of cells with DNA damage present. The mature imago group seemed to be the most aligned. Only H-type GO 0.2 treated crickets presented higher total DNA damage than the control.

The differences between strains were observed in larva and young imago groups. In larvae from the control and GO 0.02 groups, total DNA damage was higher in the D strain than in the H type. In young imago, all groups from the D strain (control, GO 0.02, and GO 0.2) presented significantly higher total DNA damage than wild-type insects.

### 2.2. DNA Damage: Double Strand Breaks (DSB)

The level of DSB was higher in larva and young imago than in mature imago groups in both examined strains (Figure 2). Statistically significant differences were noticed in some cases, and strongly outlined trends were visible in others. Larva stage groups showed higher DSB in GO-treated groups compared to the control group. In the H strain, the higher the GO concentration was, the higher DSB occurred (with significant differences between control and GO 0.2 groups). Additionally, D-strain larvae treated with GO had a higher percentage of DSB cells than the control, although these differences were not statistically significant.

In young imago, the level of DSB in H-type insects was similar between experimental groups. However, the significant differences between the control and GO 0.2 treated group were confirmed in the D strain (higher percentage of DSB in GO-intoxicated groups than in control). Mature imago insects presented relatively low DSB percentages in both strains and all experimental groups.

The comparison of strains showed a significant difference in one examined group of larvae (GO 0.02 group) with higher DSB in the long-living strain. Additionally, in young imago, the GO groups presented significantly higher DSB in the D strain than in the H. It is worth noticing that there were no differences between strains in the mature imago groups.

### 2.3. DNA Damage: 8-hydroxy-2′-deoxyguanosine (8-OHdG)

Free radical-induced oxidative lesions in DNA were also measured in the project by assessing 8-hydroxy-2′-deoxyguanosine (8-OHdG). As in previously described parameters, the most pronounced changes in DNA stability were noticed in both strains’ larva and young imago groups (Figure 3).

In the H larvae group, statistically significant differences were observed. The higher the GO concentration, the higher the 8-OHdG level was observed. On the other hand, in the D strain, the higher the GO concentration was, the lower the 8-OHdG level that was measured. In young imago groups in the wild type, both GO groups showed statistically significant differences compared to the control (the DNA damage level was much higher than in the control). There were no significant differences in the D strain in young imago groups. The mature imago groups presented evenly low 8-OHdG levels in the H and D strains. Moreover, the level of this parameter was below the detection limit in GO-treated individuals from the D strain.

The statistically significant differences between strains were marked in control and GO 0.2 groups of larvae. Moreover, in the young imago group, the differences between strains were noticed only in the GO 0.02 group showing lower DNA lesions in the D strain than in the H strain.

### 2.4. DNA Damage: Apurinic/Apyrimidinic Sites (AP Sites)

In the case of apurinic/apyrimidinic sites lesions in DNA, there were distinct differences between strains, especially in the larvae and young imago (Figure 4). The H larvae and young imago presented high levels of AP sites in almost every experimental group, compared to the D strain. In larva groups from the H strain, the level of AP sites was generally high, like in the control group. However, in the GO 0.02 group from this strain, there was a statistically lower AP site lesion level compared to the control. On the other hand, in the D strain, the results for the controls, GO 0.02 and GO 0.2, were at a low and similar level. In young imago groups, the general level of AP sites was still higher in the H strain than in the D strain. GO 0.02 group in strain H presented significantly higher AP site content when compared to the control. In mature imago, both the H and the D strains showed high AP site level with no statistical differences between experimental groups.

The comparison of strains revealed differences in most of the examined groups. The larvae groups from the H strain presented higher levels of AP sites than the D strain, with statistically confirmed differences in the control and GO 0.2 groups. In young imago, the differences between strains were significant in all experimental groups. In mature imago, the AP sites were higher in individuals from the D strain compared to the H strain. However, significant between-strain differences were confirmed only for the GO 0.02 group.

### 2.5. DNA Damage Response: Ataxia Telangiectasia Mutated (ATM) Kinase Phosphorylation (pATM)

The percentage of cells with activated ATM kinase was relatively comparable at almost all investigated insects’ life stages (Figure 5).

In the larvae group in the H strain, both GO groups had lower levels of phosphorylated ATM (pATM) than the control group (however, only for GO 0.02 group was the difference significant). Moreover, the level of pATM was higher in GO 0.2 group than in GO 0.02, but no statistically significant differences were observed. All larva groups from the D strain presented similar levels of pATM. In young imago, there was a trend in H-type strain: the higher the GO concentration was, the higher the pATM. The young imago presented similar ATM phosphorylation levels in the D strain in all measured groups. Both strains showed no statistically significant differences between experimental groups in the mature imago stage. However, there was a tendency in the D strain: the higher the GO concentration was, the greater the phosphorylation of ATM kinase that occurred.

Comparison of investigated strains showed differences only in young imago and mature imago in the GO 0.2 groups.

### 2.6. DNA Damage Response: Phosphorylation of Histone H2A.X (pH2A.X)

The percentage of cells with phosphorylation of histone H2A.X in larvae groups in wild-type insects was similar, with a slightly increased tendency in GO groups (Figure 6). The pH2A.X in the D strain was similar and comparable to wild-type insects. In the young imago group, the phosphorylated histone H2A.X in H-type crickets was equal and lower than in larvae. In the D strain, there were no significant differences—the content of pH2A.X cells was comparable between control, GO 0.02, and GO 0.2 and on the same level as in larvae groups. In the mature imago group in H-type crickets, significant differences were observed. Compared to the control, the GO 0.2 group presented a higher level of phosphorylated histone H2A.X. The GO 0.02 group also presented a higher level of pH2A.X than the control, but the difference was not statistically significant. The content of cells with activated histone in the D strain was lower in mature imago than in other age groups, but with no significant differences.

The significant differences between the strains were observed only in the young imago control (higher pH2A.X in D strain), and mature imago GO 0.2 (higher in H strain).

### 2.7. Global DNA Methylation

There were no differences between the strains in the percentage of global methylation (Figure 7). DNA methylation levels were comparable in almost all investigated groups. The methylation level in the wild type in the larvae groups was slightly higher in the GO 0.02 group. In the long-living strain, the percentage of methylated DNA was comparable in the control and GO 0.02, with a slightly higher level in the GO 0.2 group. In young imago, all wild-type groups had a similar level of DNA methylation. Additionally, in the long-living groups, there were no statistically significant differences between the experimental groups. Only GO 0.02 presented a slightly higher content of methylated DNA. Mature imago tended to show lower DNA methylation levels when exposed to GO. The tendency was more pronounced in D-strain insects, where GO-treated groups revealed significantly lower levels of DNA methylation than the control one.

## 3. Discussion

Among several theories describing the mechanism of GO action in the cell, the leading one is the concept of increased oxidative stress and its consequences [18,25,35,36,37]. Therefore, we consider the assessment of the importance of oxidative stress under specific conditions, taking into account the susceptibility to stress induced by GO in critical periods of the organism’s life. Next, we try to interpret the GO stress in the context of the specific characteristics of the organism resulting from selection for longevity.

In the following subsections, three sets of alternative hypotheses are discussed and verified:

**H1.0:** 
*The level of DNA damage and intensity of selected elements of the DNA repair pathway are similar at different life stages of the organism. The organism’s reaction to GO may depend mainly on dose and/or species-specific abilities to maintain DNA stability, rather than on the life stage.*


**H1.1:** 
*The level of DNA damage and/or efficiency of selected parameters of the DNA repair pathway differ for various life stages. The reason may be due to the diverse functions of the organism in different life stages. Intensive growth, preparation for reproduction (larvae), and reproduction (young imago) can be associated with intensive hyperphagia, an increase in metabolism and oxygen consumption. The consequence may be a different (lower/higher) susceptibility of the DNA to damage and a different (reduced/increased) ability to repair the damage caused by GO.*


**H2.0:** 
*Selection for longevity does not significantly affect the level of DNA damage, also caused by an additional stress factor. This hypothesis may be related to the supposition that life extension is not associated with DNA stability in A. domesticus, and tested components of the DNA repair pathway are also irrelevant to the potential lifespan of individuals. Both strains (wild and long-living) react similarly to GO intoxication.*


**H2.1:** 
*Both strains (wild and long-living) react differently to GO-spiked food. Assuming oxidative stress underlies the mechanism of GO toxicity and changes with aging, there are premises to believe that long-living insects deal with GO differently (better/worse). Thus, A. domesticus selected for longevity may tolerate a different (lower/higher) level of DNA damage, and the elements of the DNA repair pathway may be of varying importance to them.*


**H3.0:** 
*Global DNA methylation is similar for each developmental stage and/or strain. Due to the common phenomenon of transcriptional inactivation of chromatin, especially during growth and development, this hypothesis is unlikely. Therefore, the potential lack of differences in global DNA methylation among stages/strains/experimental groups can conclude that the parameter is not a good marker of possible changes, also due to exposure to GO. Thus, it should be assumed that the DNA methylation phenomenon is exceptionally complex, and further research should focus on assessing the methylation of selected genes.*


**H3.1:** 
*Global DNA methylation levels vary according to the developmental stage and/or strain and/or exposure to GO. Despite the expected low specificity of this parameter, it is possible to observe general relationships and use this parameter in explaining the GO toxicity mechanism in the organism.*


### 3.1. GO-Induced DNA Damage and Developmental Stage

Previous studies show that carbon-based materials (such as GO, reduced GO, nanodiamonds, carbon black, carbon nanotubes, fullerenes, carbon nanofibers, etc.) can have many adverse effects on various organisms. The reported impacts concern a wide range of measured parameters, including impairments in development/reproduction processes, cell cycles, various metabolic pathways, and lipid and protein peroxidation increases. An important symptom is also an increase in DNA damage (Table 3).

Our previous studies with the model species *A. domesticus* provided a lot of evidence for the involvement of oxidative stress in the mechanism of the toxic effect of GO. In the first step, we had shown GO to exert deleterious effects when both injected or given by oral route in the short-term (10-days), with exposure to relatively high concentrations (20 or 200 μg g^−1^) in food. In both cases, we observed increased DNA damage, enhanced apoptosis, and degenerative changes in the gut cells [13,14]. Then, we prolonged exposure time for the whole lifespan. This experiment also showed increased apoptosis, disturbed energy budget, increased DNA damage, and an array of other changes that confirmed the toxicity of GO [15]. In the third step of our research, we focused on lower concentrations of GO in food (0.2, 2, and 20mg·kg^–1^) and increased exposure time for three generations. These relatively low doses were tolerated well in the 1^st^ and 2^nd^ generation. Surprisingly, however, in the third generation, the amount of DNA damage in GO-treated groups was significantly higher than in the control [16]. This result encouraged us to reflect on the significance of this phenomenon for a population that, under chronic (multi-generational) stress conditions, may benefit from a temporary reduction in DNA stability. Consequently, it may allow for the emergence of more variants of genotypes, which, during selection, may provide better adaptation [16,38,39,40]. The mentioned above research proved the potentiality of GO to damage DNA, showing the complexity of this process, influenced by general rules and principles of physiology, genetics, and evolution. Notably, studies have always been carried out on adults (mature imago). Meanwhile, the sensitivity of organisms at earlier stages of development is yet unknown.
ijms-24-00290-t003_Table 3Table 3Cellular and tissue toxicity of different carbon nanoparticle types.Nanoparticle TypeBiological SystemConcentration/DoseFinding/EffectReferenceGraphene oxide;Graphene oxide quantum dots*M. aeruginosa*49.32 mg/L22.46 mg/LHigher ROS; increase in malondialdehyde (MDA) concentration; antioxidant enzymes disruption[41]Graphene oxide*C. elegans*0.01; 0.1 mg/L1; 10; 100 mg/LGO exposure induced autophagy in a dose dependent manner[42]Graphene oxide*C. elegans*1 μg/LMulti-generational toxicity: shortened life-span; smaller body size; reduced oocytes numbers; impairments of locomotion-related neurons[43]Graphene oxideReduced graphene oxide (rGO)*D. rerio*hepatocytes1–100 µg/mLHigher ROS in cells exposed to rGO and stopping cell replication. Nevertheless, GO did not stop cell replication, but exposed cells had higher levels of apoptosis and necrosis[44]Graphene oxideMouse2 mg/kg 5 mg/kgIncrease in peroxidase activity and MDA concentration; liver inflammation; brain—neuronal cells not affected[45]Graphene oxide*A. domesticus*0.2 mg/kg20 mg/kgIncrease in antioxidative enzyme catalase activity, lower viability of gut cells, DNA damage in hemolymph and changes in the pattern of vitellogenin protein production[31]Nanodiamonds*A. domesticus*20 µg/g200 µg/gDNA damage; increased activity of oxidative stress enzymes: catalase, glutathione peroxidase; increased total antioxidant capacity, increased level of heat shock protein[46]Ultrafine carbon blackMouse5–55 µg/mL<15 μg/mL no significant effect on splenocytes; >15 μg/mL induced ROS and malonaldehyde activity; decreased activity of superoxide dismutase and catalase; formation of protein corona in α-amylase and lipase[47]Multi-walled carbon nanotubesRat5000 mg/kgMild periarteriolar lymphoid cell depletion in the spleen; mild tubular cell degeneration on the cortex and medulla of both kidneys[48]Multi-walled carbon nanotubes*X. tropicalis*0.5 mg/L2.5 mg/LAffected the formation of spermatogonia and oocytes; no effect on the heart or liver; changed the microbial community structure and diversity of gut microbiota[49]Multi-walled carbon nanotubes*C. carpio*10 μg/L50 μg/LDownregulated expression of key steroidogenic and transcription factor genes related to testis and brain; decreases in serum testosterone and 11-ketotestosterone levels; increased activity of glutathione-S-transferases, superoxide dismutase, and catalase in both testis and brain[50]Fullerenes*M. salmoides*0.5 ppmLipid peroxidation in brain; depletion of total glutathione[51]Carbon nanofibers*P. expansa*1 mg/L10 mg/LHigher nitrite, hydrogen peroxide and lipid peroxidation levels in liver and brain; increased total glutathione, catalase and superoxide dismutase; damage in erythrocyte DNA; higher apoptosis and necrosis in erythrocytes; increased cerebral and hepatic acetylcholinesterase[52]Carbon nanofibers*Diamesa sp.,**D. cryptomeria; G. suifunensis*100 mg/LCNFs accumulated in the intestines; no toxic effects[53]Carbon nanofibers*E. fetida, D. rerio,**O. niloticus*500 μg/g10 μg/mLErythrocyte nuclear abnormalities; nanoparticle accumulation at trophic levels[54]


The analysis of the results obtained in this study revealed a significant impact of the developmental stage and GO treatment on the level of DNA damage (Table 1). Generally, neglecting strain differences (discussed below), the larvae and young imago had more significant total DNA damage, including DSB and oxidative damage (Figure 1, Figure 2 and Figure 3). The developmental stage effect was magnified (though not in every strain) by GO treatment. This means that younger individuals were more susceptible to the adverse effects of GO. The additivity of both factors was demonstrated by the insignificant interaction effect between treatment × stage. Interestingly, however, when the interaction of all three factors was examined, it turned out to be statistically significant. This result should be understood as follows: generally, GO treatment and the young age of insects are associated with higher levels of DNA damage (additive factors); however, the factor Strain can significantly modify these effects (significant interaction effect of all factors). This means that different populations of organisms, even of the same species, depending on preadaptation/selection, may not react to GO intoxication or can increase or even decrease DNA damage levels (Table 1). Undoubtedly, the higher level of DNA damage in larvae and young imago should be linked with greater energy demand and, therefore, greater oxygen consumption. The last larval instar and the young imago are characterized by hyperphagia. This ensures the energy supply for the final molting and, in the case of young imago, for reproduction. These critical stages in the animal’s life are associated with increased locomotor activity (mating behavior, stridulating) and egg production, and thus increased metabolism and oxygen consumption [55,56,57]. It was shown in the example of *Apis mellifera*, that increased activity (and oxygen consumption) was the cause of the imbalance between pro- and antioxidants and the intensification of oxidative stress, leading to an increase in DNA oxidative damage, manifested, among other things, by the rise in the level of 8-OHdG [58]. Thus, any additional stress factor based on the intensification of oxidative stress (as in the case of GO) can potentially exacerbate the adverse effects, even at relatively low concentrations.

Interestingly, AP sites tended to be higher in mature imago, and a significant difference appeared in strain D (Figure 4). AP site lesions represent the most common types of DNA damage that, when unrepaired, can disturb replication and transcription and cause mutations. In general, the mechanism of the formation of AP sites and the consequences of their occurrence have been relatively well described, especially for mammals. AP sites can be formed when removing damaged or modified bases during the base excision repair (BER) pathway [59,60,61,62]. However, the knowledge about the dynamics of changes in the number of AP sites with age is vestigial. The most recent and intriguing studies by Cai et al. describe AP sites in different tissues of mice of different ages (between 3 to 22 months old) [62]. Interestingly, the correlation of AP sites with age was tissue dependent. While correlations were negative for bone marrow, PBMC, sperm, and heart tissue, they were positive for brain and liver tissue. The authors rightly point out that their discovery can, to some extent, contradict general concepts of aging and the accumulation of damage with age [63,64,65,66,67]. As the authors also stated, science is still at the beginning of understanding the importance of AP sites in light of various biological processes, including aging [62]. Considering the above, our result allows us to conclude that in insects, or at least in *A. domesticus*, the number of AP sites does not decrease with age, and it even (in the case of long-living strain) tends to increase. However, the additional burden with GO did not affect this parameter and did not appear to be related to the mechanism of AP site formation (Figure 4).

The supply of GO also had no evident effect on DNA damage response parameters, namely ATM and H2A.X. However, the importance of the factors of treatment and stage for the pH2A.X variable was on the edge of significance. The interactions between treatment and stage factors were not significant, indicating a lack of any apparent influence of insect age on GO treatment response (Figure 5 and Figure 6; Table 2). One of the functions of ATM, a member of the PI3-kinase family, is participation in the DNA repair process (mainly DSB). Activation of this protein leads to the activation of proteins involved in the repair of damage (including H2A.X.) or the arrest of the cell cycle, with further consequences and possible induction of apoptosis. Declining ATM levels with age and deficits in ATM signaling are associated with the development of neurodegenerative diseases [68].

It has been shown that silica nanoparticles can increase the phosphorylation of ATM and other proteins, leading to apoptosis and spermatogenesis dysfunction in mice [22]. ZnO NPs can also cause oogenesis disorders with a simultaneous impairment of H2A.X activation [69]. Silver nanoparticles and TiO_2_ can cause DNA damage and modify DNA repair pathways in human cells, leading to impairment of DNA repair and/or cell cycle arrest [70,71,72]. GO can also activate ATM, but the relationship is not clear. For example, Krasteva et al., studying the effect of GO on two different colon cancer cell lines, noticed ATM activation depending on the cell type and exposure time [73]. Exposing Colon26 cells to GO did not significantly change ATM expression compared to the control. In contrast, exposure of HT24 cells to GO initially led to a significant increase (24 h of exposure) and then to an inhibition (72 h of exposure) of ATM expression [73].

The results of our research entitle us to partially accept the H1.1 hypothesis, which states that DNA damage is significantly related to stage and exposure to GO. In terms of DNA damage response parameters, we are inclined to accept hypothesis H1.0, which assumes that these parameters are not significantly related to age/stage and exposure to GO. Stage-dependent DNA damage, especially DSB and oxidative damage (Figure 1, Figure 2 and Figure 3), with the simultaneous lack or subtle activation of DNA damage response parameters (Figure 5 and Figure 6), may result from the life strategy of insects. *Acheta domesticus*, being predominantly fast-living and fast-breeding organisms, can minimize energy-demanding repair mechanisms. However, this concept should be carefully tested.

### 3.2. Selection for longevity and susceptibility to DNA damage caused by GO

Although it was noted in the previous section that the effective repair mechanisms in the case of *A. domesticus* might be of lesser importance due to its relatively short life, in this section, a closer look at this concept is presented, analyzing in particular the results for the long-living strain.

The phenomenon of prolonged life of some insects and the associated, often different, response to additional stress has been studied using several laboratory model organisms [74,75]. Moreover, data from research on social species such as bees, wasps, ants, and termites, where the queen tends to live much longer than the rest of the population, are valuable complementary material [58,75,76]. In general, it is believed that life extension is associated with increased resistance to stressors (including oxidative stress), manifested, among others, by overexpression of antioxidant enzymes [74,75]. However, some data do not fully comply with this rule (see review by Jemielity et al. [76]). In *Reticulitermes speratus* termites, the effects of UV irradiation in queens and workers were compared. It was found that the level of 8-OHdG, as well as protein and lipid peroxidation in queens, was lower than in workers after irradiation. At the same time, queens showed higher levels of catalase (CAT), and peroxiredoxin (Prx) compared to workers [77]. Our previous studies on *A. domesticus* show that lines D and H (without additional stress load) do not differ in terms of basal CAT activity and total antioxidant capacity (TAC) and, at the same time, have a higher percentage of cells with reactive oxygen species (ROS+), and cells manifesting signs of apoptosis and/or autophagy [34]. This observation supports the conclusion that a high concentration of CAT may not be significant for extending the life span of D-strain individuals. Similarly, increased expression of superoxide dismutase (SOD) was not crucial in the evolution of a long lifespan in some social insects [76].

In this study, the main effects analysis revealed that the strain factor significantly impacted the level of DNA damage but had no effect on DNA damage response parameters. The interaction between strain × treatment demonstrated the additivity of the effects on DNA damage parameters. Moreover, the pH2A.X parameter was significantly modified by the strain × treatment and strain × stage interactions (Table 1 and Table 2). This means that populations with different selection histories may respond differently regarding DNA damage and DNA damage response parameters. Undoubtedly, such a result makes it difficult to draw general conclusions, but simultaneously, it opens new perspectives for understanding the mechanism of GO action in organisms.

Total DNA damage and DSB were higher in strain D, especially in larvae and young imago (Figure 1 and Figure 2). Interestingly, the number of AP sites showed a different trend (Figure 4), and at this stage of the research, it is difficult to explain why strain D, in the early developmental stages, has fewer AP sites. The amount of 8-OHdG in the D-strain larvae was higher than in the H-type larvae. Notably, it decreased significantly after the additional stress factor (GO) was applied (Figure 3). Such a picture, again, does not allow for unambiguous verification of the hypotheses. We are inclined to accept the H2.1 hypothesis in terms of DNA damage, assuming that wild and long-living insects respond differently to the additional stress resulting from GO intoxication. Especially in the earlier stages of life, the D strain reacts more intensely (total DNA damage and DSB) or differently/inversely (AP sites and 8-OHdG) than the H strain. Regarding DNA damage response parameters, we are inclined to accept hypothesis H2.0, assuming that longevity selection does not significantly influence these parameters. However, the significantly higher level of H2A.X phosphorylation in H-strain adults, appearing after GO treatment, provides the basis for further research and more in-depth exploration of the DNA repair pathways in both strains. Moreover, proteins and pathways involved in longevity and stress resistance should be studied in both strains. They are namely the sirtuin family of proteins, IlS (the insulin/insulin-like growth factor) signaling pathway, and the target of rapamycin (TOR) pathway [75,78,79]. The elements of these pathways likely differ between individuals of the D and H strains. Therefore, their life expectancy and ability to deal with the additional stress factors (e.g., GO) vary. Our team will explore this issue in further studies.

### 3.3. Global DNA Methylation in Acheta domesticus

DNA methylation plays a vital role in gene regulation in animals. Unlike vertebrate genomes, which are methylated globally up to 60–90% (of all CpG nucleotides in mammals) [80,81], methylation in insects is targeted to genes [82]. DNA methylation was empirically detected in many insect species [82,83,84], including a few Ortoptheran species such as *Gryllotalpa fossor*, *Schistocerca gregaria*, and *Acheta domesticus* [85,86,87]. In the genome of *Acheta domesticus*, the ‘toolkit’ for DNA methylation was detected. Methyl-CpG-binding domain proteins (MBDs) are present, but there are no data about the presence of DNA methyltransferases (DNMTs) [82]. In some Orthoptera, the DNMTs were detected [88], suggesting that the house cricket is also equipped with those enzymes. MBDs are motifs that allow the selective binding of methylated DNA [89]. DNMTs are enzymes divided into three subfamilies: DNMT1—important for maintaining proper methylation status after the replication process; DNMT3—which can methylate DNA de novo; and DNMT2—which is no longer considered as a real methyltransferase [84]. In vertebrates, DNA methylation is one of the best-studied epigenetic mechanisms that lead to chromatin remodeling and changes in gene expression [90]. Chatterjee et al. revealed that GO could cause DNA damage and alter gene expression by modifying DNA methylation patterns in human bronchial epithelial cells [91]. The researchers paid attention to genes essential for maintaining the methylation status: *DNMT3B* and *MBD1*, which were hypomethylated due to GO intoxication [91]. Although the methylation process is well understood in vertebrates, the role of DNA methylation in insects remains unclear.

Based on the obtained results, the research hypothesis was adopted that global DNA methylation is similar for each developmental stage and/or strain. After long-term GO intoxication there were no significant differences in global DNA methylation in different groups of crickets. This interesting finding corresponds with results obtained from another invertebrate organism *Enchytraeus crypticus* [92]. *Enchytraeus crypticus* was intoxicated with copper oxide nanomaterials over five generations. There were no differences between the control and treated groups in the F1 DNA methylation pattern. Moreover, the statistically important differences in DNA methylation appeared in the F3-F5 generations. The authors indicated that RNAi, and not methylation, was probably the potential epigenetic mechanism affecting multigenerational exposure to metal stress [92]. Bewik et al. suggested that alternative DNA methylation pathways may exist [93]. In insects, DNMT-1 positively correlates with methylation and may act like DNMT-3. It is worth investigating more broadly the role of DNMTs, in terms of DNA methylation in insects. In the presented study, there were no differences in methylation between the developmental stages of *Acheta domesticus*. There was an assumption that DNMTs may be important regulators of longevity/aging in invertebrates [94]. For instance, the life span in fruit flies was reduced when exogenous DNMT-1 and DNMT1A were overexpressed [95]. On the other hand, DNMT2 overexpressed in *Drosophila* was essential for normal life and enhanced lifespan [96]. Kausar and Cui observed that any change in DNMTs expression might influence insects’ adaptation to different environmental conditions (for instance, extreme temperatures) by changing the DNA methylation pattern [94]. Based on the examples given, it may be supposed that perhaps the even level of methylation in strains/stages of *A. domesticus* that was observed would change in the next generations. Additionally, the role of DNMTs in insect methylation should be further explored to answer the question of how epigenetic mechanisms regulate insects’ genomes to increase flexibility and phenotypic plasticity, especially under additional chronic stress.

## 4. Materials and Methods

### 4.1. Graphene Oxide Characteristics

Graphene oxide was purchased from Nanografi, USA (graphene oxide dispersion in water; 10 mg·mL^–1^) and used for the experiment in the form provided by the manufacturer. The type of GO was selected based on suspension stability, multi-method physicochemical analysis, and in vivo cytotoxicity tests. Selected nanoparticles represent mostly single-layer, transparent flakes (with a typical height of 1.0 nm) with a small number of structural defects (Figure 8). The average flake area is about 2 μm2. Moreover, the analysis of suspension stability demonstrated that the zeta potential was around −37.40 mV.

The surface analysis of the XPS technique showed the presence of carbon (68.32%), oxygen (26.63%), and small amounts of nitrogen (1.36%), silicon (0.12%), and sulfur (3.58%). The oxygen-containing groups in the structure (21.18% (C-O) and 6.57% (C=O)) were specific to GO. Moreover, cytotoxicity studies indicated that Nanografi GO showed relatively low genotoxicity to gut cells during the longer exposition. The results of the GO toxicity study, and detailed morphological, structural, and surface analysis are presented in our previous work [9], where the selected GO was coded as Sample 3 (S3).

### 4.2. Characteristics of the Species

House cricket (*Acheta domesticus*, Gryllidae, Orthoptera, Insecta) is a common species worldwide. It is easy to breed and omnivorous. The life cycle lasts about 3 to 4 months. The size of the insect’s body is small, but big enough to collect samples for research. Those benefits make *Acheta domesticus* a promising model organism [56]. For the last twenty years, the Institute of Biology, Biotechnology and Environmental Protection at the University of Silesia in Katowice has been conducting selective breeding, resulting in two unique strains of house cricket. The strains differ in their ontogenetic development [31,32,33]. In our studies, we have used the wild type crickets (H) and the long-living strain (D), which have longer ontogenetic development time and longer life expectancy compared to the wild type.

### 4.3. Food Preparation: Graphene Oxide Food, Control Food

The GO-spiked and control food was prepared using the same protocol. Briefly, the food was prepared by grinding the standard artificial food (Kanisan Q). Then it was mixed with GO dissolved in ultrapure water in two variants: 0.2 mg·kg^–1^ of dry food (GO 0.2 group) and 0.02 mg·kg^–1^ of dry food (GO 0.02 group). The food was then dried for 24 h at 45 °C, sterilized, and kept in dry conditions in sealed plastic bags. In control food, instead of the GO suspension, the same volume of ultrapure water was added, and the following procedures were the same as for GO-spiked food.

### 4.4. Experimental Model

The animals were bred in a room with standard conditions (temperature: 28.8 ± 0.88 °C; photoperiod L:D 12:12; humidity: 20–45%) optimal for an insect’s development. The larvae that were obtained from the laboratory stock population (wild type (H) and long-living type (D)) were divided into experimental groups and kept in insectaries. In total, six groups were created. Larvae were provided with water and food ad libitum for their whole life. Strains (H and D) were fed food contaminated with GO in two concentrations: 0.2 mg·kg^–1^ (GO 0.2 group) and 0.02 mg·kg^–1^ (GO 0.02 group). Control groups for each strain were fed with uncontaminated food.

At the last larva stage, before metamorphosis, ten larvae from each experimental group were collected for future analysis. Then the steps were repeated on the 20th day of imago life (young imago) and on the 30th day of the imago stage (mature imago).

### 4.5. DNA isolation

DNA was used for measuring the following: DNA global methylation, AP sites and oxidative stress biomarker 8-hydroxy-2′-deoxyguanosine (8-OHdG). DNA was isolated using Genomic Mini Kit (A&A Biotechnology, Gdansk, Poland). Briefly, RNA Later (Merck) was removed with a tube from the gut. Then tissue was homogenized, twice washed with Tris buffer (Genomic Mini Kit, A&A Biotechnology) and then all the steps in the protocol were followed. DNA was suspended in Tris buffer (100 µL), and DNA quality and concentration were measured using Nano Drop200 (Thermo Fisher Scientific, Waltham, MA USA). DNA was stored at −80 °C for future analysis.

### 4.6. Tissue Preparation for Flow Cytometry

Gut tissues were used for flow cytometry, and prepared as described elsewhere [16,31,34]. The cell suspension was treated according to the protocol supplied with the kit (Luminex, Austin, TX USA; MCH200107). Briefly, 10 µL of cells were suspended in a fixation buffer for 10 min on ice, washed, and permeabilized in a cold buffer. Then, antibody cocktail solution (phospho-specific ATM (Ser1981)-PE and a phospho-specific Histone H2AX-PECy5) was added to each tube and incubated (30 min, RT, dark). Finally, the cells were centrifuged (300× *g*, 5 min), washed in PBS (3 times), and resuspended in 200 µL of 1x Assay Buffer immediately before the measurement.

### 4.7. Measurements of Selected Parameters

#### 4.7.1. Total DNA Damage, Double Strand Breaks (DSB), pATM and pH2A.X

DNA damage in gut cells was assessed by Guava^®^ Muse Cell Analyzer through an ATM and H2A.X activation using a Muse Multi-Color DNA Damage Kit (Luminex, Austin, TX USA; MCH200107). The degree of histone phosphorylation at position Ser139 (γH2A.X) increases with DNA damage, accumulating at the DNA double-strand breaks. The ATM protein coordinates the DNA repair chain by activating relevant enzymes. The results were presented as dot plot, based on the following cell populations: pATM (−) and pH2A.X (−) cells (no DNA damage), pATM (+) and pH2A.X (−) cells (ATM activation), pATM (−) and pH2A.X (+) cells (H2A.X activation) and dual activation of ATM and H2A.X—cells with double-strand breaks. The measurements were performed as described previously [34].

#### 4.7.2. Apurinic/Apyrimidinic (AP) Sites

AP sites were measured using a DNA Damage Quantification Colorimetric Kit (BioVision, Waltham, MA USA). Briefly, DNA was defrosted on ice and diluted in TE buffer to obtain 0.1 µg∙µL^–1^ concentration. Then, the manufacturer’s protocol was executed. The number of AP sites in DNA were calculated using a standard curve.

#### 4.7.3. 8-hydroxy-2′-deoxyguanosine (8-OHdG)

8-OHdG level in the DNA of the digestive tract cells was measured using an Enzyme-linked immunosorbent assay kit DNA Damage Competitive ELISA Kit (Invitrogen). Samples were defrosted on ice and diluted to 100 ng of DNA in 50 µL of assay buffer. All measurements were performed according to protocol. The values of 8-OHdG level in DNA were calculated based on a standard curve.

#### 4.7.4. Global DNA Methylation

Global DNA methylation was measured using Imprint^®^ DNA Quantification Kit (Sigma-Aldrich, Saint Louis, MO USA). DNA was defrosted on ice and diluted. The DNA concentration in every single sample was 150 ng in 30µL of DNA binding solution. Then, the protocol was followed. To calculate methylation levels the standard curve method was used.

### 4.8. Statistical Analysis

Depending on the availability of the tissue, 3–5 samples were analyzed (n = 3–5). For AP sites, 8-OHdG, and global methylation, 4 technical repetitions were made for each sample, and the results were then averaged. Complete information on the n value in each experimental group is provided under the Figures. The results in the Figures are represented as mean ± SE. Two tests were used to check the normality: the Kolmogorov–Smirnov and Shapiro–Wilk. The homogeneity of the variances was verified by the Levene and Brown–Forsythe tests. The data fulfilled the analysis of variance criteria. Therefore, parametric tests were used. Two-way ANOVA (post-hoc Fisher test, *p* < 0.05) was applied to assess the effects of independent variables on dependent variables. Moreover, to determine the effect of all factors (stage, strain, GO treatment) and their interactions on dependent variables, MANOVA (Wilks’ Lambda test; *p* < 0.05) was used. Statistica software (STATISTICA^®^13 PL, StatSoft Inc., Tulsa, OK USA) was used for statistical analysis.

## Figures and Tables

**Figure 1 ijms-24-00290-f001:**
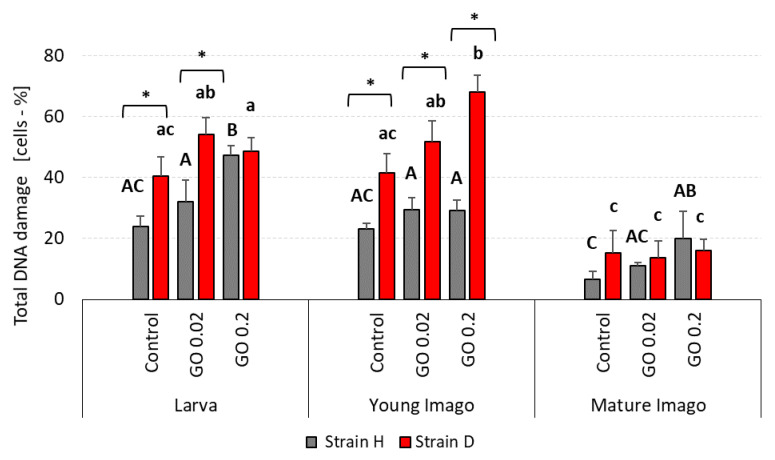
Total DNA damage in the gut cells of the wild (H) and long-living (D) strains of *A. domesticus* that had been chronically intoxicated with graphene oxide (GO). Abbreviations: Measurements were conducted at larva (n = 5), young imago (n = 5) and mature imago stage (n = 3). Control—animals fed uncontaminated food; GO 0.02 and GO 0.2 groups of animals fed GO contaminated food at a concentration of 0.02 or 0.2 mg·kg^–1^ of dry food, respectively. Significant differences were measured using ANOVA (Fisher test; *p* < 0.05). Different letters denote differences among the experimental groups in the strain. Capital letters refer to strain H, while lowercase letters refer to strain D; asterisks indicate differences between the strains in each experimental group separately.

**Figure 2 ijms-24-00290-f002:**
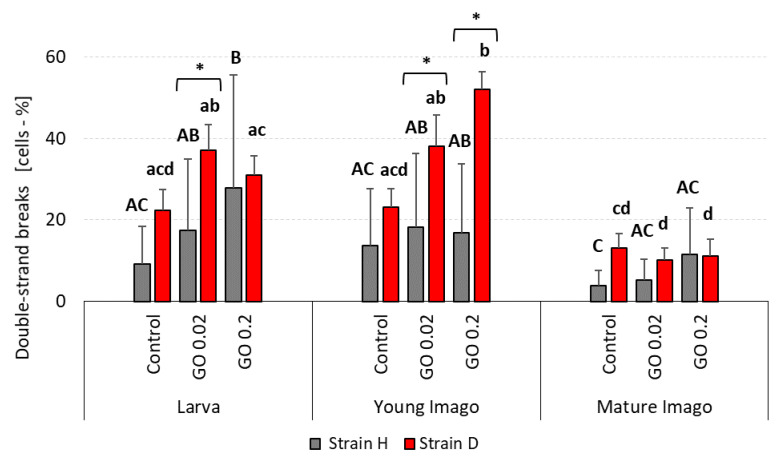
DSB (double strand breaks) in DNA in the gut cells of the wild (H) and long-living (D) strains of *A. domesticus* that had been chronically intoxicated with graphene oxide (GO). Measurements were conducted at larva (n = 5), young imago (n = 5) and mature imago stage (n = 3). Abbreviations: see Figure 1. Capital letters refer to strain H, while lowercase letters refer to strain D; asterisks indicate differences between the strains in each experimental group separately.

**Figure 3 ijms-24-00290-f003:**
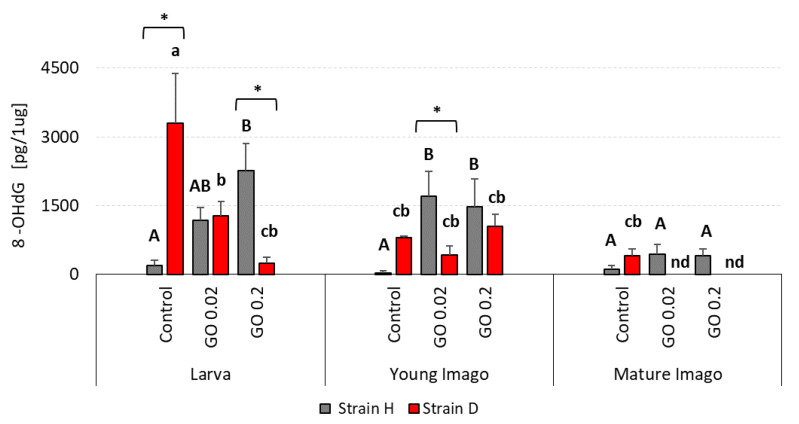
8-OHdG (8-hydroxy-2′-deoxyguanosine) lesions in DNA in the gut cells of the wild (H) and long-living (D) strains of *A. domesticus* that had been chronically intoxicated with graphene oxide (GO). Measurements were conducted at larva (n = 5), young imago (n = 5) and mature imago stage (n = 5). Abbreviations: see Figure 1. Capital letters refer to strain H, while lowercase letters refer to strain D; asterisks indicate differences between the strains in each experimental group separately.

**Figure 4 ijms-24-00290-f004:**
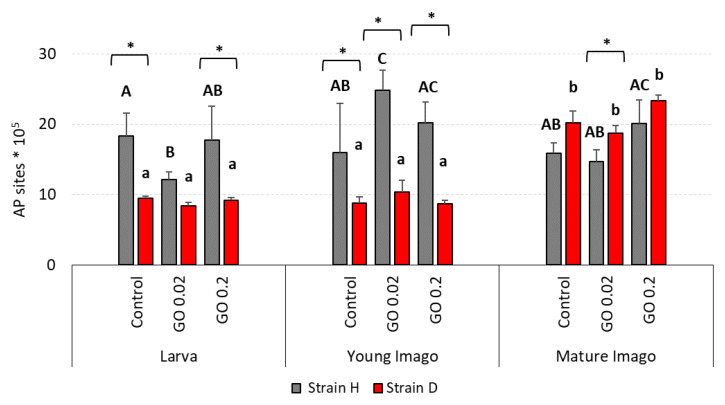
AP sites (apurinic/apyrimidinic sites) lesions in DNA in the gut cells of the wild (H) and long-living (D) strains of *A. domesticus* that had been chronically intoxicated with graphene oxide (GO). Measurements were conducted at larva (n = 4), young imago (n = 4) and mature imago stage (n = 4). Abbreviations: see Figure 1. Capital letters refer to strain H, while lowercase letters refer to strain D; asterisks indicate differences between the strains in each experimental group separately.

**Figure 5 ijms-24-00290-f005:**
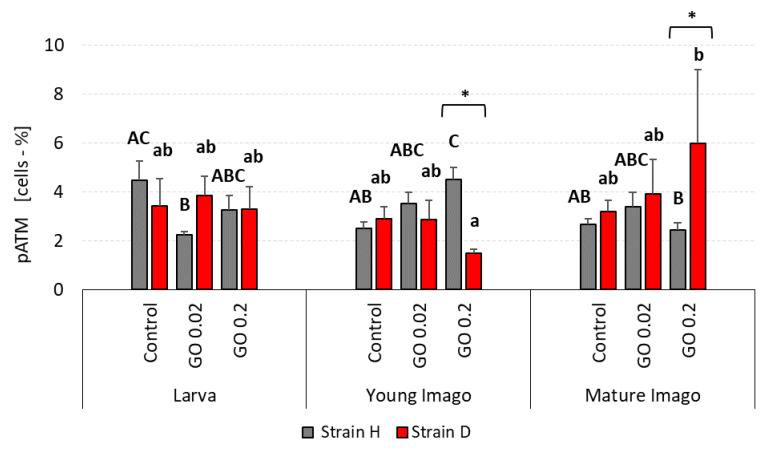
pATM (activated ATM kinase) percentage in the gut cells of the wild (H) and long-living (D) strains of *A. domesticus* that had been chronically intoxicated with graphene oxide (GO). Measurements were conducted at larva (n = 5), young imago (n = 5) and mature imago stage (n = 3). Abbreviations: see Figure 1. Capital letters refer to strain H, while lowercase letters refer to strain D; asterisks indicate differences between the strains in each experimental group separately.

**Figure 6 ijms-24-00290-f006:**
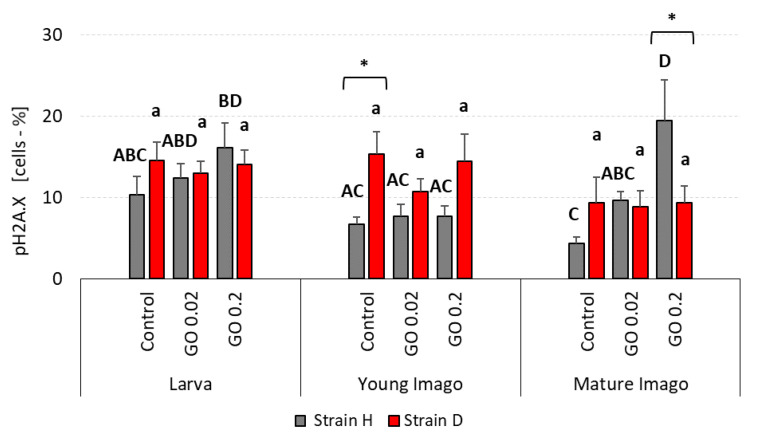
pH2A.X (phosphorylation of histone H2A.X) percentage in the gut cells of the wild (H) and long-living (D) strains of *A. domesticus* that had been chronically intoxicated with graphene oxide (GO). Measurements were conducted at larva (n = 5), young imago (n = 5) and mature imago stage (n = 3). Abbreviations: see Figure 1. Capital letters refer to strain H, while lowercase letters refer to strain D; asterisks indicate differences between the strains in each experimental group separately.

**Figure 7 ijms-24-00290-f007:**
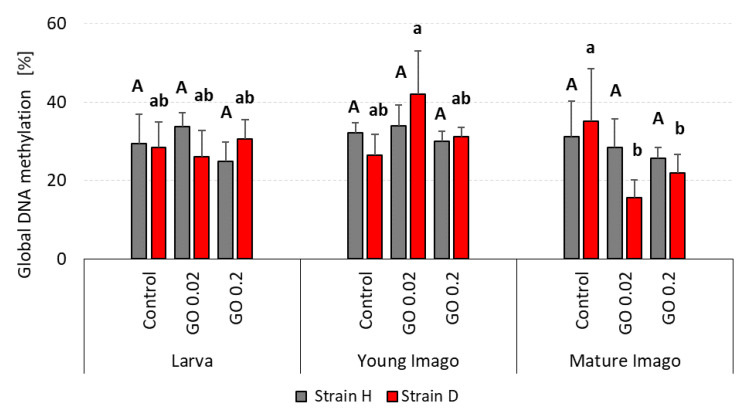
Global DNA methylation percentage in the gut cells of the wild (H) and long-living (D) strains of *A. domesticus* that had been chronically intoxicated with graphene oxide (GO). Measurements were conducted at larva (n = 5), young imago (n = 5) and mature imago stage (n = 5). Abbreviations: see Figure 1. Capital letters refer to strain H, while lowercase letters refer to strain D.

**Figure 8 ijms-24-00290-f008:**
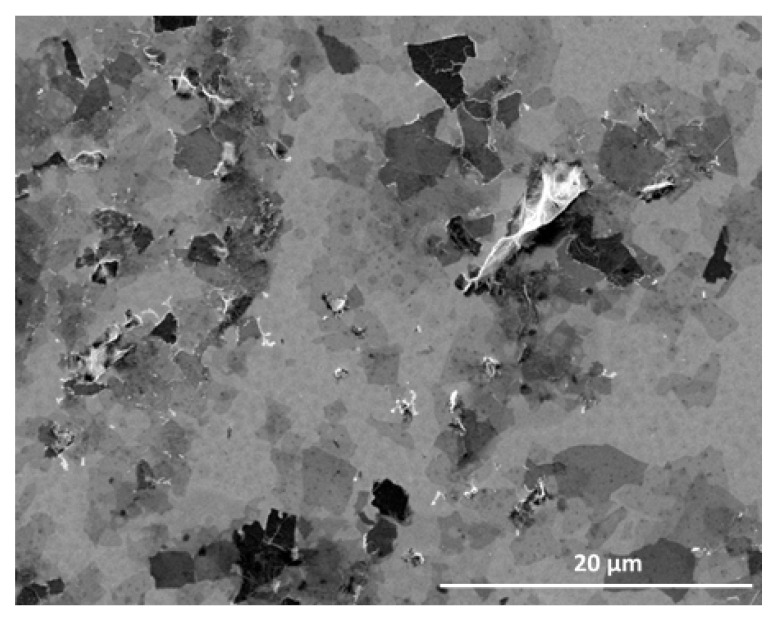
SEM image of GO. Magnification: 5000×, scale bar: 20 µm.

**Table 1 ijms-24-00290-t001:** The main effects and interactions of the factors: Strain, Treatment, and Stage on DNA damage parameters in the cells of *Acheta domesticus*.

Effects ^1^	DNA Damage Parameters ^2^	
	F	*p*
Strain (1)	9.715	<0.0001
Treatment (2)	2.514	0.0152
Stage (3)	7.858	<0.0001
(1) × (2)	1.203	0.3042
(1) × (3)	4.583	<0.0001
(2) × (3)	1.298	0.2041
(1) × (2) × (3)	1.826	0.0317

^1^ The factors (independent variables) and their interactions, i.e., mutual modifications of their impact on dependent variables (DNA damage parameters). ^2^ The individuals (larvae, young imago, and mature imago) from two strains (H—wild type and D—long-living type) were exposed to graphene oxide (GO) in food in two different concentrations (0.2 and 0.02 mg∙kg^−1^ of food) during the whole lifetime. Tested variables: Total DNA damage, DSB—double strand breaks, 8-OHdG—8-hydroxy-2′-deoxyguanosine, AP sites—apurinic/apyrimidinic site (MANOVA, Wilks’ Lambda test).

**Table 2 ijms-24-00290-t002:** The main effects and interactions of the factors: Strain, Treatment, and Stage on DNA damage response parameters in the cells of *Acheta domesticus*.

Effects ^1^	DNA Damage Response Parameters ^2^			
	pATM		pH2A.X	
	F	*p*	F	*p*
Strain (1)	0.237	0.6284	1.863	0.1774
Treatment (2)	0.147	0.8638	3.049	0.0548
Stage (3)	0.726	0.4882	3.020	0.0563
(1) × (2)	0.117	0.8900	3.300	0.0437
(1) × (3)	2.570	0.0849	3.583	0.0339
(2) × (3)	0.665	0.6190	0.935	0.4496
(1) × (2) × (3)	2.010	0.1044	0.957	0.4378

^1,2^ Abbreviations: as in Table 1. Tested variables: pATM—phosphorylated ataxia telangiectasia mutated (ATM) kinase, pH2A.X—phosphorylated H2A.X histone.

## Data Availability

Raw data are provided on the RepOD database (accession: https://doi.org/10.18150/8UCJCA).

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
