# Peer review of "Age- and Lifespan-Dependent Differences in GO Caused DNA Damage in *Acheta domesticus"

_ijms, 2022, doi:10.3390/ijms24010290_

Round 1
Reviewer 1 Report
Review comments for ijms-2030094 entitled “Age- and lifespan-dependent differences in GO caused DNA damage in Acheta domesticus” described by Barbara Flasz et al.
This study investigated DNA damages induced by graphene oxide (GO) toxicity using Acheta domesticus. In past years, studies have always been carried out on mature imago; the authors investigated the sensitivity of GO at earlier stages of development and exhibited GO toxicity is depended on the life stages and lifespan. This paper is interesting and will impact the related research area, such as nanotoxicology—however, some minor changes are required for the reader’s understanding.
1. The hypotheses H1.0 to H3.0 in the introduction should be organized more straightforwardly to relationship experiments and discussion. How about these hypotheses move to the top of the discussion section?
2. In Tables 1 and 2, the multiple classification analysis procedures could be unclear. There needs to be a description of the definition of each effect and interaction.
3. In Figures 1 to 7, the labels such as A and ab need to be explained what that means.
4. Although the shape of GO is exhibited, average particle size and surface chemical composition need to be clarified. The distribution of GO in larvae and imago is also unknown.
5. Sample size is relatively small (n = 3~5). Is it enough to guarantee the reproducibility of experiments and correct statistical analysis?
Reviewer 2 Report
The work presented by Flasz et al. presents the effects of GO in different life stages of the organism Acheta domesticus. The article is interesting, the introduction is very well written, and it presents very good results, but some points need to be improved so that this article can be published:
11. Tables 1 and 2 present interesting results, but are poorly explored. Please improve the discussion regarding these results.
22. Nomenclatures must be unified once they are already abbreviated. For example, graphene oxide on line 166.
33. Some gaps were missing for my understanding. I believe that exposure to graphene (using food as an entryway) was done continuously. If so, the authors should perform discontinuous exposure experiments in order to observe whether the DNA damage is altered.
44. In the discussion section a table should be added showing the toxic action of carbon-based materials (such as carbon black, carbon nanofibers, fullerenes, graphite, graphene, carbon nanotubes, etc.) on other organisms, summarizing the results and exposing damage correlations (not just DNA) of these materials to the organisms used. This will help to reinforce the raised hypotheses, once comparisons with other types of materials are made.
10.1093/toxsci/kfp265
10.33594/000000573
10.1016/j.cbi.2019.04.036
10.1007/978-0-387-76713-0_13
10.3390/app12020720
10.1080/10408444.2017.1391746
10.3109/15376516.2012.754534
10.3390/biomedicines9091155
10.1002/smll.201201417
10.33594/000000382
10.1166/jbn.2011.1224
10.1016/j.envres.2018.05.027
Etc…
